# Geochemistry and Geometrical Features of the Upper Cretaceous Vitulano Para-Autochthonous Karst Bauxites (Campania Region, Southern Italy): Constraints on Genesis and Deposition

Roberto Buccione [1],*[ID], Stefano Vitale [2][ID], Sabatino Ciarcia [3][ID] and Giovanni Mongelli [1][ID]

[1] Department of Sciences, University of Basilicata, 85100 Potenza, Italy; giovanni.mongelli@unibas.it
[2] Department of Earth Science, Environment and Resources (DiSTAR), University of Naples Federico II, 80126 Naples, Italy; stefano.vitale@unina.it
[3] Department of Sciences and Technologies, University of Sannio, 82100 Benevento, Italy; sabatino.ciarcia@unisannio.it
* Correspondence: roberto.buccione@unibas.it

**Abstract:** In the Vitulano area, Upper Cretaceous bauxite bodies fill small depressions and karst cavities within Cretaceous shallow-water limestones. These bauxites were studied to understand the processes that led to their formation. Geochemical, mineralogical, and petrographic analyses were carried out on the bauxite samples, together with image analysis providing geometric parameters. The texture of Vitulano bauxite consists of ooids and sub-circular aggregates dispersed in a predominantly Ca-rich matrix. Ooids are generally formed by a single large core, often surrounded by an alternation of different aggregates of boehmite and Al-hematite reflecting different climate periods. The composition is dominated by the major elements $Al_2O_3$ and $CaO$ with lower concentrations of $Fe_2O_3$ and $SiO_2$. Boehmite, calcite, hematite, and anatase are the main mineralogical phases identified. Image analysis provided values of fractal dimension D that gives information on carbonate platform exposure times since it is linked to long-lasting sub-aerial events and diffusion-limited cluster aggregation processes. The tectonic evolution of the area played an important role in the genesis of the Vitulano bauxites since it favored the erosion, transport, and re-deposition of pre-existing bauxite material from the surrounding Campania bauxites. Based on this hypothesis, Vitulano bauxites are defined and classified as para-autochthonous, and this was supported by $Eu/Eu*$ vs. $Sm/Nd$ and $Eu/Eu*$ vs. $TiO_2$ $Al_2O_3$ indices displaying a similarity between Vitulano and the other Campania bauxites.

**Keywords:** geochemistry; image analysis; Campania bauxites; southern Italy; para-autochthonous deposit

## 1. Introduction

Bauxites are residual rocks that form in a subaerial environment in regions characterized by tropical and subtropical climates [1–3]. The interest in bauxite ores has been growing because they are not only a major source of aluminum, but also a useful indicator for paleoclimatic and paleogeographic reconstructions [4–6]. The chemical weathering processes occurring during bauxite formation may lead to the enrichment of several chemical elements, such as Al, Ti, Fe, and some trace metals [7,8], especially high field strength elements and also to the leaching of mobile elements [9].

In order to redefine the processes and paleo-conditions that led to the formation of bauxites, geochemical element discrimination and heavy mineral tracing methods have been widely used. These models investigate the distribution of weathering-resistant elements in bauxites, such as Ti, Zr, Nb, Ta, and Rare Earth Elements (hereafter REEs) as a tool to unroof parental affinity by comparing the abundances of immobile elements in both bauxite deposits and potential source rocks [10–18]. Great interest has been focused on the

Mediterranean Bauxite Province [1,19] since this bauxite province is closely related to the tectonic evolution and paleoclimatic conditions of the Tethyan realm [20,21].

In southern Italy, several bauxite deposits occur, such as the autochthonous Cretaceous and the allochthonous Cenozoic karst-type bauxite deposits [11,22] consisting of multiple bauxite layers formed within the Cretaceous shallow-water carbonate platform sequence [20]. The Cenozoic bauxites are classified as Salento-type deposits, formed by the erosion and re-deposition of pre-existing Cretaceous bauxite rocks [22]. Previous studies on Italian bauxite deposits focused on ore deposition processes [19], mineralogy [23], elemental geochemistry [8,11,17,22,24,25], and zircon age for paleogeographic restoration [5,18,26]. These studies provided constraints for the provenance of the southern Italy bauxite deposits, the main mineralogical composition, the fractionation of REEs and other critical raw materials, and paleoenvironmental–paleoclimatic conditions.

This study focuses on patchy bauxite occurrence filling karst cavities and fractures hosted within Cenomanian–Coniacian radiolitid limestones located in the western sector of Mt. Camposauro, around the town of Vitulano (Campania region, southern Italy) [27] (Figure 1). In addition to elemental geochemistry, mineralogy, and petrography, fractal geometry and image analysis were used to better address the bauxitization process [5,24]. In the Campania region, studied bauxite deposits were limited to the Matese Mts. and the Caserta district; hence, the discovery and analytical characterization of these bauxite bodies may provide new analytical data and constraints for the genesis, provenance, and paleo-condition of their formation.

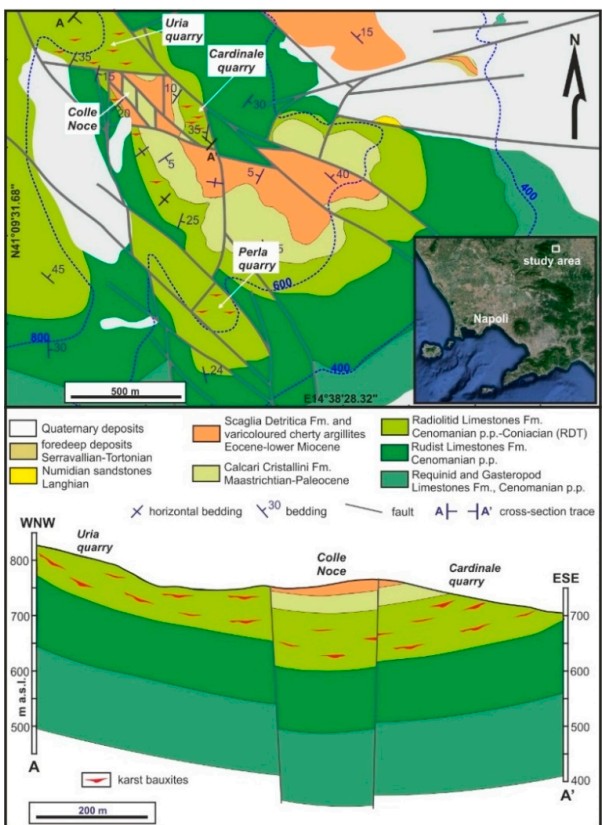

**Figure 1.** Geological map and cross section of the study area (modified after [28]).

## 2. Geological Framework

The southern Apennines consist of the tectonic superposition of several thrust sheets belonging to different paleogeographic domains, including oceanic successions (Ligurian units [28]) and shallow-water to pelagic sedimentary deposits of the continental part of the down-going Adria plate (Apennine Platform, Lagonegro–Molise Basin, and Apulian

Platform) [29,30]. The rocks hosting the studied bauxites exposed at Mt Camposauro belong to the Apennine Platform domain. The pre-orogenic succession consists of Triassic–Upper Cretaceous shallow-water carbonates directly covered by Miocene synorogenic rocks. However, in the SE sector (Figure 1) where the karst bauxites are exposed, the succession upward passes to margin-to-slope deposits, including Maastrichtian–Paleocene recrystallized limestones and an Eocene–Lower Miocene argillitic successions (scaglia-type deposits).

Within the Cenomanian–Coniacian Radiolitid Limestones Fm., bauxite deposits fill karst cavities (Figure 2a) and fractures (Figure 2b). They are formed by whitish and reddish calcarenites and calcilutites, hosting fragments and ooids of bauxites (Figure 2a). Karst cavity walls are frequently covered by calcite cementation (Figure 2c). Normally, bauxite deposits are dissected by normal and reverse faults (Figure 2c,d) locally showing the syn-depositional characters. These structures with opposite kinematics are interpreted as related to collapse [31]), in this case associated with karst processes [27]. According to Vitale et al. [28], the bauxitic sedimentation was synchronous with an extensional event that occurred during an abortive rifting that affected the northern Africa sector [32] and started in the Albian–Cenomanian interval. This extensional episode caused the dismembering of the Apennine Platform with the passage of several sectors from shallow water to margin and slope conditions; in contrast, other carbonate blocks passed to sub-aerial conditions. This is the case of the Matese Mts., including Mt. Camposauro, where the continental exposure allowed the formation of bauxite deposits (Figure 2) and a pervasive karsism with the formation of several cavities filled by the bauxite sediments.

These bauxites have been quarried for a long time for use as ornamental stones called Vitulano and Cautano Marbles [33]. Examples of their use can be found in numerous churches in the Campania region and the royal palaces of Caserta and Naples.

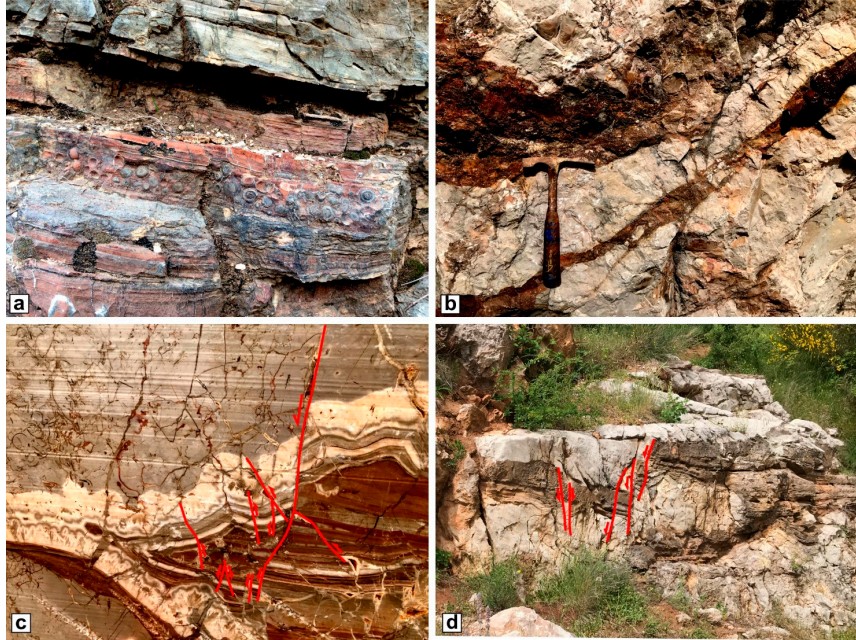

**Figure 2.** Examples of bauxite occurrence in the study area: (**a**) karst cavity filled by calcareous deposits hosting bauxitic pisolites (Colle Noce); (**b**) fractures filled by bauxite (Uria quarry); (**c**) karst cavity filled by calcite cement and bauxite material (Uria quarry); (**d**) parallel-bedding karst cavity filled by bauxite, hosting syn-sedimentary faults (Uria quarry). Red lines represent the faults, whereas the red arrows indicate the sense of shear.

## 3. Sampling and Analytical Methods

Fifteen samples were taken from karstified Radiolitid Limestones Fm. outcrops in the Uria quarry, where bauxite fills several small karst cavities and fractures.

Petrographic and microstructural observations were performed by using a Nikon Alphaphot-2 YS2 optical microscope at the Department of Sciences, University of Basilicata.

The mineralogical composition was determined by X-ray powder diffraction (XRPD) at the Department of Sciences, University of Basilicata, Italy, using a Siemens D5000 powder diffractometer with Cu-K$\alpha$ radiation and a 40 kV, 32 mA, and 0.02° (2$\theta$) step size setup. Samples were first milled in a Retsch planetary mill equipped with two agate jars and agate milling balls to generate a very fine powder. Microchemical and micromorphological analysis was made with scanning electron microscopy (ESEM) and an XL30 Philips LaB6 ESEM instrument equipped with an energy dispersive X-ray spectrometer (SEM–EDS) at the Microscopy Laboratory of the Department of Sciences, University of Basilicata, Italy.

Major oxides and trace element abundances were determined by ICP and ICP-MS analysis at Activation Laboratories (Ancaster, Canada) after sample powders were digested using a four-acid attack (HF, HClO$_4$, HNO$_3$, and HCl). A sample of 0.25 g was firstly digested using hydrofluoric acid, then with a mixture of nitric and perchloric acids, before being heated in several ramping and holding cycles using precise programmer-controlled heating that took the samples to incipient dryness. After incipient dryness was attained, the samples were brought back into solution using aqua regia before being analysed using Varian ICP and PerkinElmer Sciex ELAN 9000 ICP-MS instruments. Total loss on ignition (LOI) values was gravimetrically estimated after overnight heating at 950 °C.

Image analysis was performed on bauxite sample images by using the free software ImageJ which is a Java-based image processing software. The software provided the binarisation of each sample image, which produced black (ooids) and white (bauxite matrix) images. The fractal dimension D measurement was made using the automatic box-counting technique that splits the sample image into several boxes of decreasing sizes and measures the number of boxes filled by ooids (filling frequency) in each dimensional class. The fractal dimension D value is the slope of the straight line obtained from fitting the pairs of values of the natural logarithm of the filling frequency vs. the natural logarithm of the box size.

## 4. Results

### 4.1. Texture, Mineralogy, and Geometric Parameters

The texture of karst bauxites consists of ooids, sub-circular structures growing around a nucleus and dispersed in a fine matrix (Figure 3a,b). In the studied samples, the ooids can be monomineralic, since they consist of only one single mineralogical phase, generally boehmite or hematite, whereas in other cases, they are formed by a large nucleus surrounded by a thin layer of different mineral composition (Figure 3c,d). We also observed larger structures that incorporate numerous ooids with different shapes and compositions, which have probably undergone transport, deformation, and re-deposition (Figure 3e). The shape and the organisation of these structures are chaotic, including several aggregates showing different composition and suggesting they are the result of erosion, transport, and re-sedimentations of pre-existing bauxites, indicating, according to the definition of Bardossy [7], the Vitulano bauxite is para-autochthonous. Calcite-rich veins were found in several samples (Figure 3f).

Microchemical and micromorphological analysis (SEM-EDS) confirmed what was observed in the petrographic analysis. The ooids may either consist of a core surrounded by a layer of different composition (Figure 4a,b) or may be formed by a core surrounded by an alternation of a large number of mineral layers (Figure 4c,d). In some cases, ooids are formed by a large boehmite core surrounded by thinner hematite layers, while other ooids consist of hematite surrounded by several layers of boehmite.

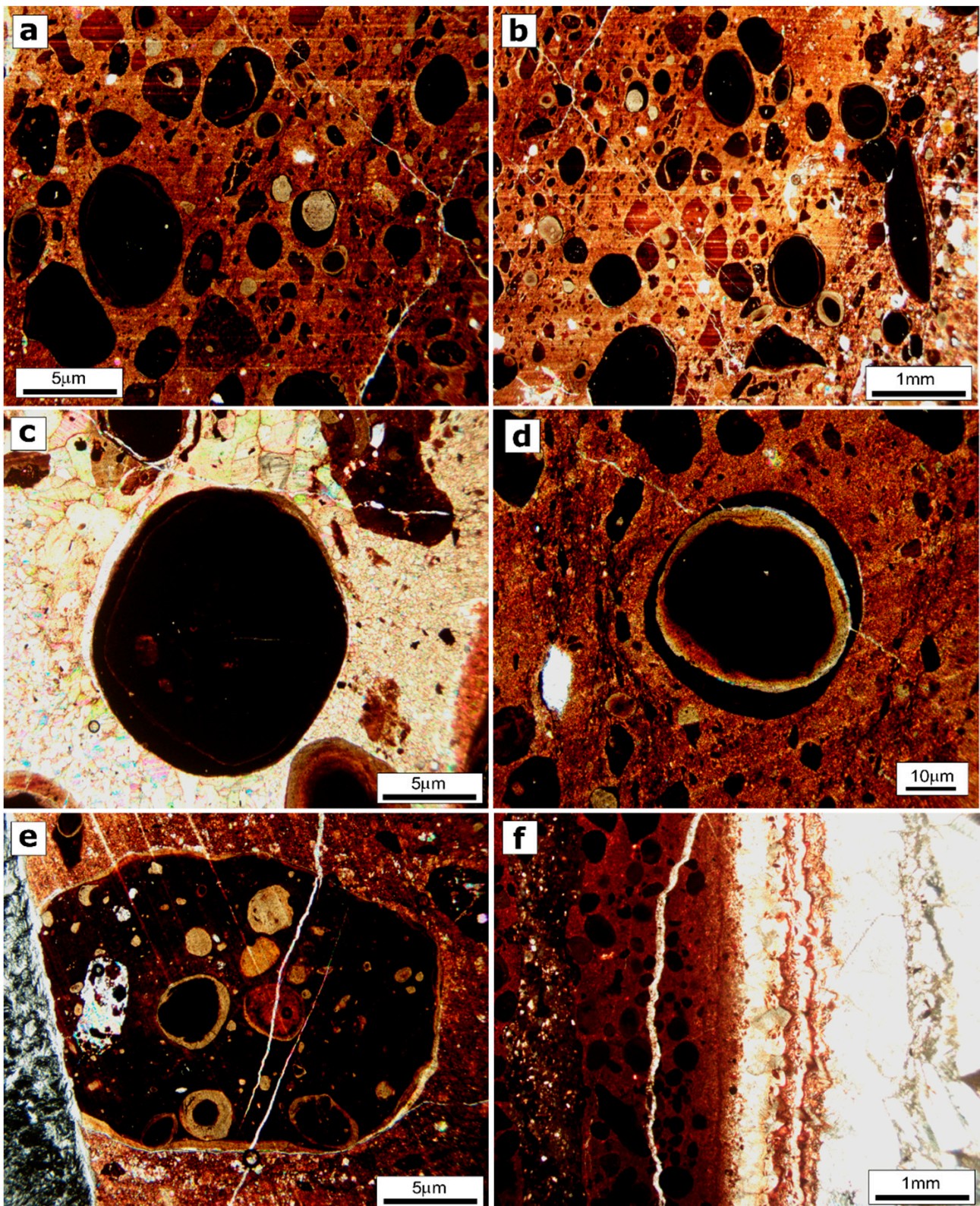

**Figure 3.** Petrographic observations of selected bauxite samples showing typical textural features: (**a**,**b**) typical texture of karst bauxites with ooids dispersed in a fine matrix; (**c**,**d**) larger sub-circular ooids with homogeneous mineral composition; (**e**) large bauxite fragment including several ooids and aggregates; (**f**) contact zone between bauxite and calcite vein.

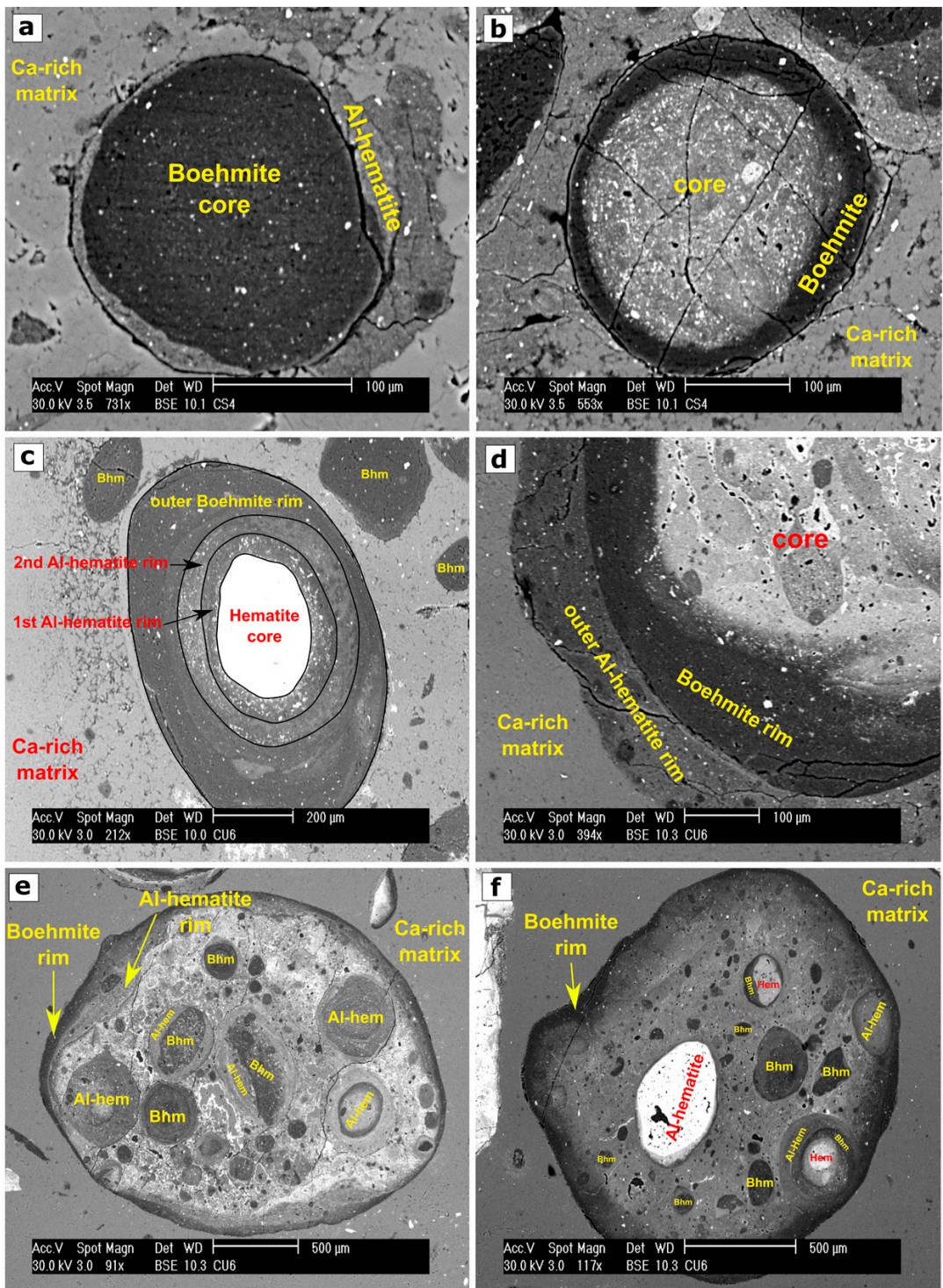

**Figure 4.** SEM-EDS images showing the typical texture of studied bauxites: (**a**) boehmite core surrounded by a thin layer of Al-hematite in a Ca-rich matrix; (**b**) Al-hematite core surrounded by a boehmite rim; (**c**) large Al-hematite core surrounded by several layers of different composition; (**d**) multi-mineral core with two layers of boehmite and Al-hematite in a Ca-rich matrix; (**e**,**f**) two larger chaotic structures incorporating several aggregates and ooids in a Ca-rich matrix.

Larger structures incorporating numerous ooids, which are likely the result of redeposition of pre-existing bauxites, were observed mainly surrounded by an outer rim of boehmite (Figure 4e,f). Accessory minerals, such as zircon and monazite, have been de-

tected in boehmite concretions. XRPD analysis revealed that the mineralogical composition of the Vitulano bauxite samples is dominated by calcite and boehmite followed by hematite and anatase (Figure 5).

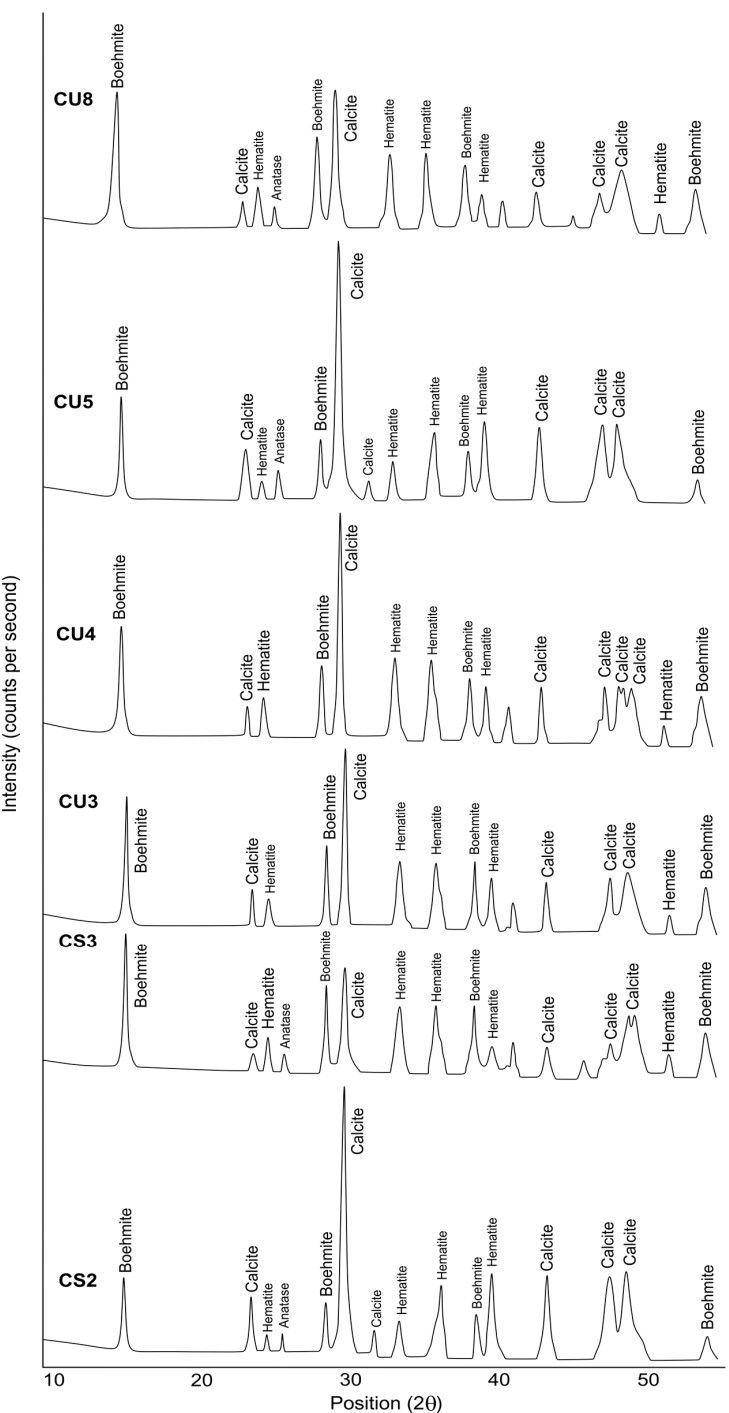

**Figure 5.** XRPD patterns of selected bauxite samples.

Regarding image analysis and geometric parameters, the average of the fractal dimension D in the Vitulano bauxite samples is D = 1.86.

### 4.2. Geochemistry

The geochemical composition of the studied bauxites is dominated by the major oxides $Al_2O_3$ (med = 27.83 wt.%) and CaO (med = 25.64 wt.%) with less abundance of $Fe_2O_3$

(med = 13.74 wt.%) and $SiO_2$ (med = 5.28 wt.%) (Figure 6, Table 1). We calculated the interquartile range (IQR) for each element which is a measure of statistical dispersion indicating the spread of the data. IQR is the difference between the third and first quartiles of the data (Q3-Q1) in the box plot and is the length of each box. For major elements, $Al_2O_3$ (IQR = 17.4) and CaO (IQR = 18.1) exhibit high variability, while $SiO_2$ (IQR = 2.1) and $Fe_2O_3$ (IQR = 10.9) show lower values of variability.

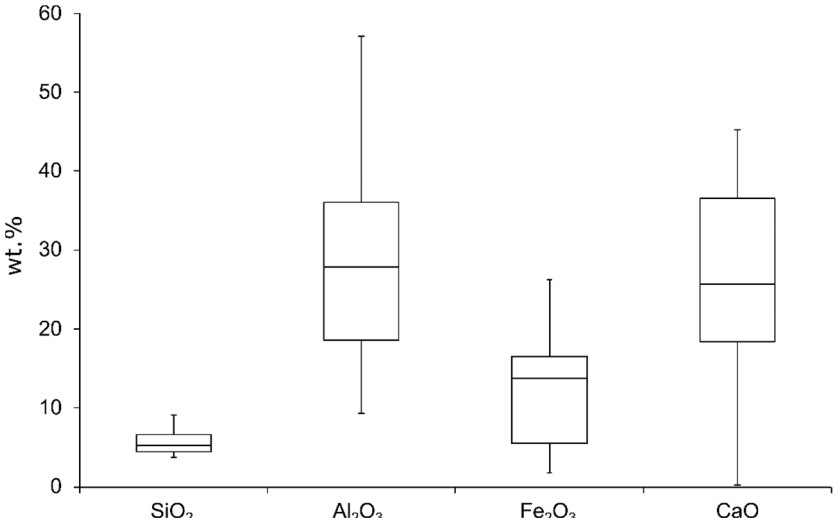

**Figure 6.** Box plot showing major oxide compositions (wt.%) of Vitulano bauxites.

The most abundant trace elements are Zr (med = 257 ppm), Cr (med = 230 ppm), Sr (med = 157 ppm), and V (med = 118 ppm), while further trace elements, such as Sc (med = 30 ppm), Ba (med = 39 ppm), Y (med = 44 ppm), Co (med = 20.5 ppm) Ni (med = 120 ppm), Cu (med = 30 ppm) (med = 118 ppm), Zn (med = 80 ppm), Ga (med = 28 ppm), Nb (med = 26 ppm), Th (med = 41 ppm), and Th (med = 20.9 ppm), show lower concentrations. High variability was observed for Cr (IQR = 190) and Zr (IQR = 158), while the rest of the trace elements, such as Ni (IQR = 80), V (IQR = 68.5), Zn (IQR = 52.5), Sr (IQR = 45), Pb (IQR = 32), Ba (IQR = 30.5), Y (IQR = 28), Sc (IQR = 19), Cu (IQR = 15), and Nb (IQR = 12.5) show from moderate to low variability (Figure 7). It has to be noted that the sample CS2 shows high Sr value (1520 ppm), and this can be explained by the presence of an Sr-rich calcite in the matrix or by isomorphogenic substitution and enrichment in $SrCO_3$.

Total REEs concentration ($\Sigma$REEs) ranges from 141.74 ppm to 715.5 ppm with a median of 356 ppm. The chondrite-normalised (Figure 8) (chondrite values are from Evensen et al. [34]) REEs pattern shows a moderate fractionation with $(La/Yb)_{cho}$, which ranges from 9.65 to 13.80 with median values of 10.64. REEs normalised patterns show both positive and slightly negative Ce anomalies with Ce/Ce* median of = 0.98, while Eu exhibits only moderately negative anomalies with an Eu/Eu* median = of 0.71 ranging from 0.68 to 0.72.

**Table 1.** Major (wt.%) and trace elements (ppm) in the Vitulano bauxite samples.

| Element wt. % | d.l. | CS1 | CS2 | CS3 | CS4 | CS5 | CS6 | CS7 | CU1 | CU2 | CU3 | CU4 | CU5 | CU6 | CU7 | CU8 |
|---|---|---|---|---|---|---|---|---|---|---|---|---|---|---|---|---|
| $SiO_2$ | 0.01 | 7.51 | 7.07 | 5.28 | 6.03 | 5.67 | 4.36 | 6.53 | 3.89 | 9.05 | 3.76 | 5.24 | 6.65 | 4.65 | 3.84 | 4.47 |
| $Al_2O_3$ | 0.01 | 56.57 | 32.66 | 41.2 | 57.1 | 26.96 | 37.86 | 16.45 | 9.26 | 9.88 | 27.83 | 28.46 | 20.75 | 22.84 | 10.94 | 34.18 |
| $Fe_2O_3$ | 0.01 | 20.34 | 12.44 | 13.74 | 20.47 | 8.15 | 26.31 | 5.54 | 2.58 | 1.77 | 15.47 | 15.56 | 3.68 | 13.97 | 5.48 | 17.41 |
| MnO | 0.001 | 0.18 | 0.11 | 0.24 | 0.18 | 0.33 | 0.38 | 0.06 | 0.03 | 0.03 | 0.13 | 0.12 | 0.04 | 0.09 | 0.04 | 0.13 |
| MgO | 0.01 | 0.58 | 0.82 | 0.55 | 0.5 | 0.61 | 0.43 | 1.01 | 0.88 | 0.57 | 0.38 | 0.51 | 0.69 | 0.57 | 0.85 | 0.55 |
| CaO | 0.01 | 0.19 | 21.62 | 16.6 | 0.44 | 28.52 | 11.14 | 37.53 | 45.19 | 41.51 | 25.64 | 24.3 | 35.49 | 29.64 | 41.86 | 20.05 |
| $Na_2O$ | 0.01 | 0.03 | 0.03 | 0.02 | 0.02 | 0.02 | 0.01 | 0.03 | 0.02 | 0.02 | 0.01 | 0.01 | 0.02 | 0.01 | 0.02 | 0.03 |
| $K_2O$ | 0.01 | 0.29 | 0.39 | 0.14 | 0.23 | 0.19 | 0.11 | 0.39 | 0.19 | 0.22 | 0.08 | 0.19 | 0.23 | 0.11 | 0.17 | 0.09 |
| $TiO_2$ | 0.001 | 2.54 | 1.50 | 1.79 | 2.64 | 1.23 | 1.76 | 0.77 | 0.47 | 0.45 | 1.25 | 1.36 | 1.08 | 1.07 | 0.52 | 1.65 |
| $P_2O_5$ | 0.01 | 0.1 | 0.28 | 0.61 | 0.08 | 0.28 | 0.1 | 0.09 | 0.1 | 0.17 | 0.06 | 0.08 | 0.13 | 0.06 | 0.08 | 0.05 |

**Table 1.** *Cont.*

| Element wt. % | d.l. | CS1 | CS2 | CS3 | CS4 | CS5 | CS6 | CS7 | CU1 | CU2 | CU3 | CU4 | CU5 | CU6 | CU7 | CU8 |
|---|---|---|---|---|---|---|---|---|---|---|---|---|---|---|---|---|
| LOI | - | 12 | 23.28 | 20.23 | 11.96 | 26.58 | 17.13 | 31.45 | 36.54 | 34.84 | 25.09 | 24.14 | 31.02 | 27.77 | 34.75 | 21.32 |
| Total | 0.01 | 100.3 | 100.2 | 100.4 | 99.63 | 98.54 | 99.6 | 99.86 | 99.16 | 98.49 | 99.71 | 99.96 | 99.76 | 100.8 | 98.57 | 99.92 |
| ppm | | | | | | | | | | | | | | | | |
| Sc | 1 | 40 | 30 | 38 | 41 | 26 | 60 | 21 | 12 | 16 | 37 | 35 | 14 | 25 | 14 | 37 |
| Be | 1 | 8 | 4 | 6 | 8 | 4 | 7 | 2 | 1 | <1 | 4 | 4 | 2 | 3 | 2 | 5 |
| V | 5 | 189 | 118 | 132 | 188 | 82 | 217 | 48 | 28 | 21 | 133 | 132 | 71 | 115 | 57 | 119 |
| Ba | 2 | 64 | 55 | 44 | 56 | 59 | 73 | 39 | 15 | 15 | 24 | 39 | 26 | 31 | 17 | 29 |
| Sr | 2 | 156 | 1520 | 164 | 145 | 198 | 238 | 184 | 166 | 133 | 149 | 107 | 291 | 114 | 147 | 157 |
| Y | 1 | 61 | 39 | 69 | 60 | 50 | 71 | 34 | 22 | 22 | 44 | 46 | 25 | 29 | 21 | 48 |
| Zr | 2 | 498 | 292 | 335 | 520 | 239 | 400 | 144 | 94 | 87 | 257 | 277 | 201 | 225 | 106 | 326 |
| Cr | 20 | 440 | 220 | 240 | 450 | 140 | 310 | 70 | 40 | 40 | 240 | 230 | 90 | 250 | 80 | 300 |
| Co | 1 | 35 | 22 | 43 | 34 | 31 | 49 | 6 | 1 | <1 | 19 | 15 | 9 | 14 | 5 | 22 |
| Ni | 20 | 200 | 120 | 190 | 200 | 150 | 200 | 140 | 40 | 40 | 110 | 110 | 80 | 100 | 50 | 150 |
| Cu | 10 | 40 | 30 | 30 | 40 | 30 | 40 | 10 | 20 | 20 | 30 | 30 | 20 | 20 | 10 | 110 |
| Zn | 30 | 110 | 80 | 110 | 120 | 80 | 110 | 40 | <30 | <30 | 60 | 80 | 50 | 60 | 30 | <30 |
| Ga | 1 | 58 | 33 | 41 | 61 | 26 | 41 | 17 | 9 | 7 | 28 | 30 | 20 | 24 | 11 | 38 |
| As | 5 | 30 | 26 | 17 | 32 | 11 | 36 | 11 | 5 | 41 | 17 | 17 | 8 | 21 | 14 | 27 |
| Rb | 2 | 17 | 16 | 6 | 13 | 8 | 5 | 15 | 7 | 8 | 4 | 10 | 9 | 6 | 8 | 6 |
| Nb | 1 | 50 | 31 | 26 | 53 | 24 | 36 | 17 | 9 | 10 | 26 | 28 | 23 | 23 | 11 | 34 |
| Mo | 2 | 4 | 4 | 7 | 5 | 5 | 6 | 2 | <2 | <2 | 3 | 2 | <2 | <2 | <2 | 3 |
| Sn | 1 | 10 | 6 | 5 | 10 | 4 | 8 | 3 | 1 | 2 | 5 | 6 | 4 | 5 | 2 | 7 |
| Sb | 0.5 | 7.5 | 6.2 | 5 | 7.6 | 3.3 | 7.2 | 2.5 | 2 | 1.7 | 10.4 | 9.7 | 4 | 9.5 | 4.6 | 8.5 |
| Cs | 0.5 | 3.8 | 4.2 | 2.2 | 3.8 | 2 | 1.9 | 3.6 | 1.5 | 1.2 | 1.5 | 2.6 | 2.7 | 2 | 1.9 | 1.8 |
| Hf | 0.2 | 12.5 | 7.7 | 3.2 | 13.8 | 6.4 | 10.9 | 3.8 | 2.3 | 2.4 | 7 | 7.3 | 5.3 | 6.2 | 2.9 | 9.4 |
| Ta | 0.1 | 1.5 | 0.9 | 0.3 | 0.9 | 0.8 | 1.1 | 1.1 | 0.6 | 0.6 | 1.2 | 1.4 | 1.2 | 1.3 | 0.7 | 2.4 |
| W | 1 | 8 | 7 | 7 | 6 | 7 | 4 | 5 | <1 | 1 | 10 | 6 | 3 | 3 | 3 | 6 |
| Tl | 0.1 | 0.3 | 0.2 | 0.2 | 0.3 | 0.3 | 0.2 | 0.1 | <0.1 | <0.1 | 0.1 | 0.3 | 0.1 | 0.2 | <0.1 | <0.1 |
| Pb | 5 | 64 | 41 | 41 | 68 | 26 | 79 | 20 | 9 | 20 | 51 | 35 | 12 | 42 | 18 | 53 |
| Th | 0.1 | 31.1 | 21.9 | 24.3 | 33.9 | 15.4 | 37.5 | 10.7 | 5.6 | 7.6 | 21.1 | 20.9 | 15.2 | 19.6 | 9.2 | 32.1 |
| U | 0.1 | 3.3 | 2.1 | 2.5 | 3.5 | 1.7 | 2.6 | 1.1 | 0.7 | 0.6 | 2.3 | 1.8 | 1.5 | 1.5 | 0.9 | 2.4 |
| La | 0.1 | 109 | 75.3 | 91.2 | 113 | 70.4 | 145 | 50.3 | 30 | 30.8 | 91.5 | 81.9 | 41.4 | 55.8 | 33.8 | 93.7 |
| Ce | 0.1 | 211 | 217 | 180 | 209 | 167 | 336 | 92.6 | 51.3 | 58.1 | 176 | 140 | 135 | 116 | 69.3 | 152 |
| Pr | 0.05 | 22.2 | 17.1 | 20.3 | 21.9 | 16.7 | 32.4 | 11.7 | 7.53 | 7.1 | 19.8 | 18 | 8.41 | 11.2 | 7.84 | 19.1 |
| Nd | 0.1 | 77.7 | 65.4 | 78.2 | 77.1 | 64.9 | 122 | 45 | 30.1 | 27.5 | 72.5 | 67.7 | 30.4 | 40.6 | 29.3 | 71 |
| Sm | 0.1 | 13.2 | 13 | 15.4 | 13.1 | 12.6 | 22.1 | 9.3 | 6 | 5.9 | 13.1 | 13.2 | 5.4 | 7.1 | 5.7 | 13.9 |
| Eu | 0.05 | 2.66 | 2.66 | 3.28 | 2.59 | 2.69 | 4.59 | 2 | 1.3 | 1.32 | 2.67 | 2.73 | 1.07 | 1.46 | 1.22 | 2.88 |
| Gd | 0.1 | 10.5 | 9.6 | 10 | 10.3 | 10.4 | 17.3 | 7.7 | 5 | 4.9 | 9.8 | 10.2 | 4.2 | 5.7 | 4.7 | 11.4 |
| Tb | 0.1 | 1.8 | 1.3 | 2.1 | 1.8 | 1.6 | 2.7 | 1.2 | 0.8 | 0.8 | 1.5 | 1.6 | 0.7 | 0.9 | 0.7 | 1.8 |
| Dy | 0.1 | 11.5 | 7.6 | 12.2 | 11.4 | 9.3 | 15.1 | 7.3 | 4.4 | 4.5 | 8.8 | 9 | 4.2 | 5.7 | 4 | 10.7 |
| Ho | 0.1 | 2.3 | 1.5 | 2.4 | 2.3 | 1.7 | 2.8 | 1.3 | 0.8 | 0.8 | 1.7 | 1.7 | 0.9 | 1.1 | 0.8 | 2.1 |
| Er | 0.1 | 6.4 | 4 | 6.3 | 6.6 | 4.7 | 7.3 | 3.4 | 2.1 | 2.2 | 4.4 | 4.6 | 2.6 | 3.2 | 2 | 6.1 |
| Tm | 0.05 | 1.03 | 0.63 | 0.95 | 1.04 | 0.69 | 1.11 | 0.51 | 0.31 | 0.32 | 0.69 | 0.69 | 0.41 | 0.5 | 0.3 | 0.88 |
| Yb | 0.1 | 6.9 | 4.3 | 6.4 | 7.2 | 4.5 | 7.1 | 3.4 | 2.1 | 2.1 | 4.4 | 4.7 | 2.8 | 3.3 | 2 | 5.6 |
| Lu | 0.01 | 1.08 | 0.69 | 0.98 | 1.16 | 0.7 | 1.08 | 0.56 | 0.34 | 0.34 | 0.68 | 0.74 | 0.47 | 0.52 | 0.32 | 0.79 |
| ΣREEs | | 477.27 | 420.08 | 432.71 | 478.49 | 367.88 | 716.58 | 236.27 | 142.08 | 146.68 | 407.54 | 356.76 | 237.96 | 253.08 | 161.98 | 391.95 |

Note: d.l.: detection limit; Eu/Eu * = [Eu$_{cho}$/$\sqrt{(Sm_{cho} \times Gd_{cho})}$]; Ce/Ce * = [Ce$_{cho}$/$\sqrt{(La_{cho} \times Pr_{cho})}$]; <: indicates that the element composition is lower than the detection limit.

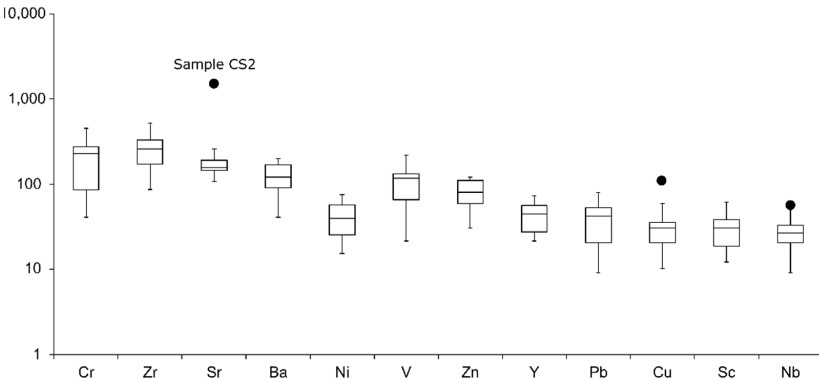

**Figure 7.** Box plot showing trace element (ppm) compositions of Vitulano bauxites. Black circles indicate outliers.

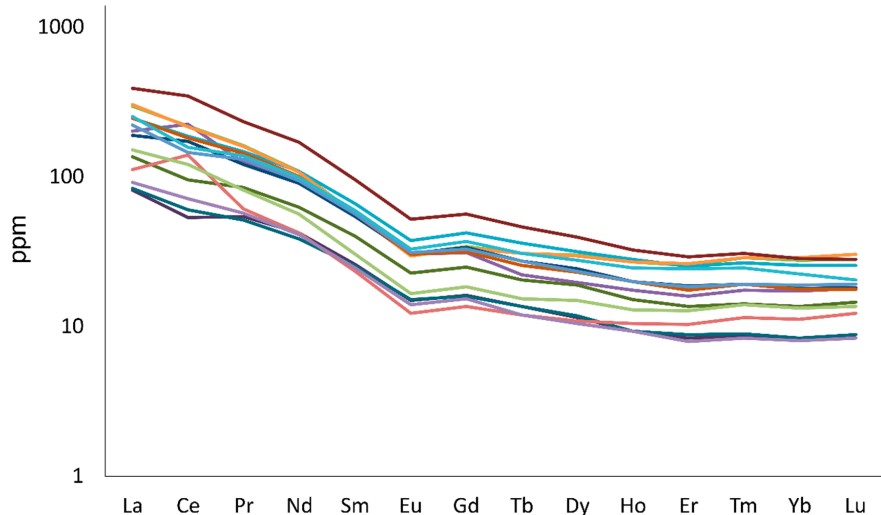

**Figure 8.** Chondrite-normalised REEs patterns of studied bauxites.

## 5. Discussion

### 5.1. Image Analysis in Bauxites

Image analysis represents a valuable approach for analysing the properties of a different range of structures ranging from molecules and atoms to the coastlines of continents [35], and the aggregation of small particles to form larger structures, such as the formation of minerals, can be described in terms of fractal geometry [36]. The growth of sub-circular concretions, such as ooids within bauxites, is a chemical transport-controlled process explained with diffusion-based models [37–41] and can be described as the growth of fractal aggregates using a molecular diffusion pattern [36,42].

Previous studies on southern Italy karst bauxites performed image analysis on textural components, providing some geometric parameters of the ooids, such as circularity, aspect ratio, and fractal dimension (D) [5,24]. For instance, the fractal dimension average (D) for the Salento autochthonous bauxite pebbles is very close to D values related to diffusion-limited aggregation models, while Campania bauxites exhibit higher fractal dimension values [24], which are close to the fractal dimension associated with the diffusion-limited cluster aggregation processes where small particles join together to form further clusters that continue to keep on joining to form larger and larger clusters [36].

The average fractal dimension D in the Vitulano bauxite samples (median = 1.86; Figure 9) is very similar to the values of the Matese Mts. and Caserta district bauxites (median = 1.85). Since it has been demonstrated that the fractal dimension D provides information on carbonate platform exposure during bauxite deposition [5,24], this finding suggests that Campania bauxite deposits formed, overall, during the same emersion span of the Apennine carbonate platform, such as the Matese Mts and the Caserta district [30,32]. This is consistent with the genesis of the Vitulano bauxites, which, in fact, derives from the alteration, transport, and deposition of pre-existing bauxites.

### 5.2. Genesis of Vitulano Bauxites

The Upper Cretaceous to Eocene evolution of the Vitulano area was characterized by a NW–SE extension due to a rifting event [30,32]. In the Maastrichtian–Paleocene interval, some blocks locally emerged and started to experience erosion, whereas others sank and were characterized by the deposition of margin-slope carbonates ("Calcari Cristallini" Fm.). In the Eocene–Miocene period, the subsidence continued with the deposition of deep basin clayey and calcareous sediments ("Scaglia Detritica" Fm.).

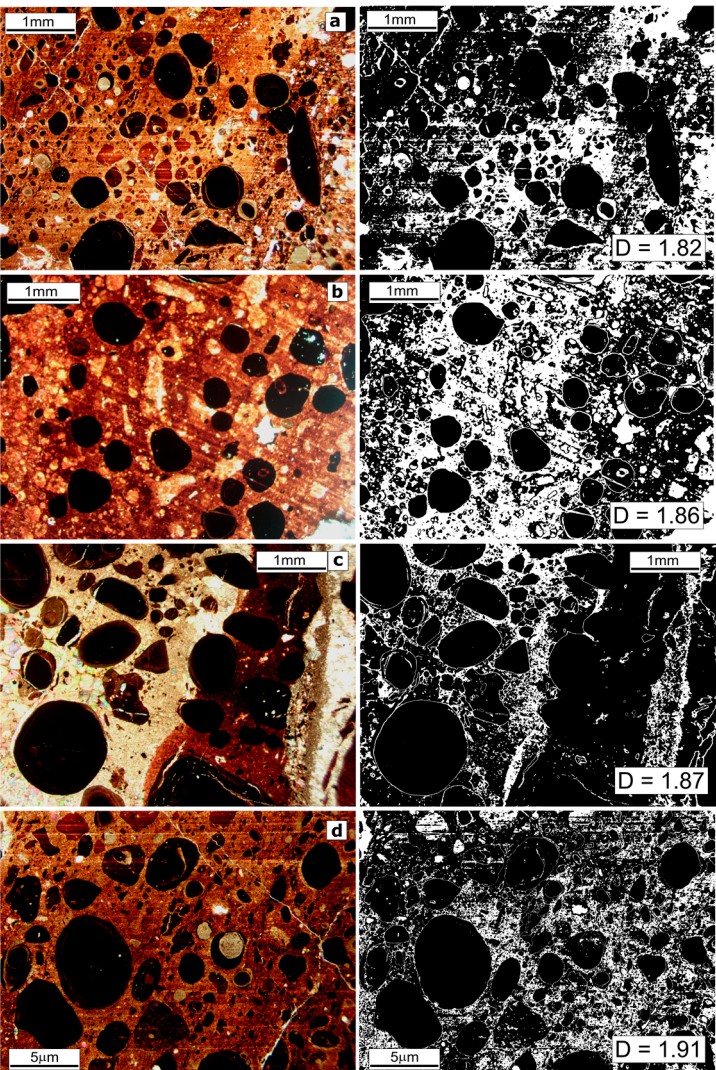

**Figure 9.** Binary images of selected samples for image analysis. The binarization of sample images was provided by the software Image J which produced black (ooids) and white (bauxite matrix) images. (**a**) binary image of the sample CS2; (**b**) binary image of the sample CS4, (**c**) binary image of the sample CU1, (**d**) binary image of the sample CU3.

The age of the Vitulano bauxites was dated back to the Cenomanian–Coniacian [30,32]; this could indicate that the bauxites that outcrop extensively in the Matese Mts. and Caserta district areas [26,43–45] may have undergone weathering processes that led to the formation of bauxites under study. These processes of dismantling, transport, re-sedimentation, and diagenesis probably took place during the first phase of the evolution of the Vitulano area, which includes, in fact, an extension event characterized by the emergence of limestone rocks and pre-existing bauxite deposits.

The composition and textural features of karst bauxites result from several processes, such as parent-rock(s) dissolution, neo-mineralization, sedimentation, transport, and diagenesis [11,22,26].

The main process that allowed the sediment to evolve with its typical appearance and structure is karstic activity, which can create micro-fractures, more or less extensive, pervasive fractures, and karstic voids in the carbonate substrate that can subsequently be filled by bauxite [22,24].

The studied bauxites do not share the same features observed in the Matese Mts. and the Caserta district deposits consisting of flat, contiguous lenses with a thickness of a few meters. Rather they occur as small tasks and depressions filled with bauxitic

material. The karst cavities in the Vitulano area are filled, as clearly indicated by the texture features, by a para-autochtonous bauxite that, in turn, is affected by post-depositional micro-fractures filled by calcite, due to the precipitation of calcium carbonate from circulating solutions. Consequently, the Vitulano bauxite, different from what has been observed in other southern Apennine deposits [7] and consistent with the geochemical and mineralogical composition, shows high CaO abundances and is calcite-rich.

Further, and similarly to what was observed in the other southern Italy bauxites, the alternation of Al-rich and Fe-rich concretions would be conditioned by climatic factors: hematite cores indicate periods in a humid tropical environment, while boehmite formed under conditions of low water activity in climatic regimes characterized by reduced rainfall [22,24].

The alternation of different climatic conditions is also suggested by the ooid/matrix ratio and the geometric features of the ooids characterized by a hematite core surrounded by alternating accretions of boehmite and hematite.

The positive Ce anomaly in some samples reflects the peculiar redox chemistry of Ce and can be due to the $Ce^{3+}$ to $Ce^{4+}$ oxidation with the precipitation of cerianite ($CeO_2$) or by its further dissolution followed by the formation of Ce-rich fluorocarbonate minerals, as observed in other upper Cretaceous karst bauxite deposits in southern Italy [11]. All samples show uniformly negative Eu anomalies which are usually associated with mafic source rocks and furthermore reveal that the bauxite formation is not affected by the Eu anomaly values in the studied bauxites [8].

In the Vitulano area, the presence of normal faults and fractures bounding the karst cavities, together with new fractures cutting off the bauxite deposits and filled with the same sediments, support the presence of an active extensional regime that entered the platform through major faults [27]. This scenario suggests a different sedimentological evolution for the various blocks that experienced differential uplift and subsidence).

In order to confirm the Vitulano bauxite derived from pre-existing Cenomanian–Turonian bauxite bodies occurring in the Matese Mts. and the Caserta district, we used geochemical indices, including Eu/Eu *, Sm/Nd, and $TiO_2/Al_2O_3$ that have proven their effectiveness in assessing parental affinity of karst bauxites [11]. The Eu/Eu * vs. Sm/Nd and Eu/Eu * vs. $TiO_2/Al_2O_3$ binary plots (Figures 10 and 11) strongly suggest the Vitulano bauxites are closely related to the Albian–Cenomanian bauxites of the Campania Region, confirming portions of the latter were affected by dismantling and transport, thus originating the para-autochtonous Vitulano bauxite.

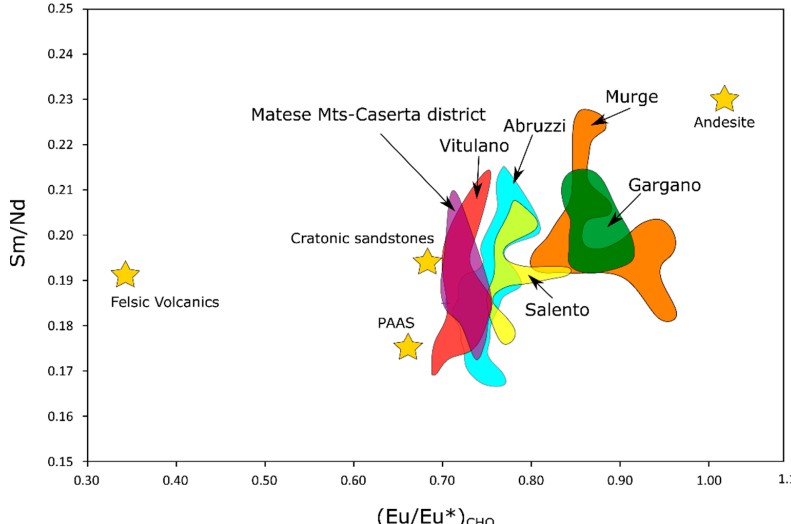

**Figure 10.** Eu/Eu * vs. Sm/Nd binary diagram showing field composition of Italian bauxites [5,7,11,22,24–26]. Felsic rocks and cratonic sandstones values are from [46]; PAAS (Post-Archean Australian Shales) and andesite values are from [47]; limestone values are from [11].

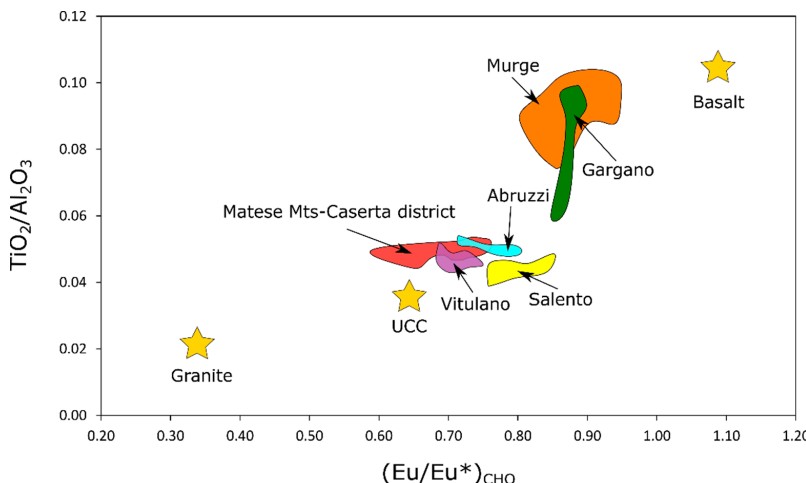

**Figure 11.** Eu/Eu* vs. $TiO_2/Al_2O_3$ binary diagram showing field composition of Italian bauxites [5,7,11,22,24–26]. UCC (upper continental crust [47]).

## 6. Conclusions

i. In the environs of the town of Vitulano, several small bauxite bodies have been found, appearing as filling of small depressions and karst cavities on which minero-chemical and petrographic assessments were performed together with image analysis assessment.

ii. The petrographic analysis revealed that the texture of the studied bauxites consists of sub-circular aggregates (ooids) dispersed in a fine Ca-rich matrix.

iii. The mineralogy of aggregates mainly consists of boehmite and Al-hematite, reflecting different climate conditions that occurred during their formation since boehmite formed in a dry climate, while Al-hematite formed during more humid periods.

iv. The chemical composition of Vitulano bauxites is dominated by $Al_2O_3$ and CaO and subordinately by $Fe_2O_3$ and $SiO_2$.

v. Tectonic activity has controlled the formation and deposition of the studied bauxites, which are para-autochthonous, as demonstrated by their textural and compositional features, deriving through erosion, transport, and the re-deposition of pre-existing bauxitic deposits.

vi. Textural and parental affinity geochemical indices, together with image analysis parameters, indicate that the original deposits that formed the Vitulano bauxites most likely were exposed in the Matese Mts. and the Caserta district areas which are located close to the studied bauxites.

**Author Contributions:** Conceptualization, R.B. and G.M.; sampling activity, R.B., S.V., S.C. and G.M.; laboratory analysis and methodology, R.B. and G.M.; writing—original draft preparation, R.B.; supervision, G.M., S.V. and S.C. All authors have read and agreed to the published version of the manuscript.

**Funding:** This research received no external funding.

**Institutional Review Board Statement:** Not applicable.

**Informed Consent Statement:** Not applicable.

**Data Availability Statement:** Not applicable.

**Acknowledgments:** Authors are indebted to the owner of Cava Uria, arch. Gennaro Esposito, who helped us for the sampling activity of the studied bauxites. In addition, authors want to thank editor and reviewers for their helpful comments and suggestions that improved the final version of the manuscript.

**Conflicts of Interest:** The authors declare no conflict of interest.

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
