# Peer review of "Geochemistry and Geometrical Features of the Upper Cretaceous Vitulano Para-Autochthonous Karst Bauxites (Campania Region, Southern Italy): Constraints on Genesis and Deposition"

_minerals, doi:10.3390/min13030386_

Round 1

Reviewer 1 Report

Geochemistry and geometrical features of the Vitulano  para-autochthonous karst bauxites (Campania region, southern Italy): constraints on genesis and deposition” by Buccione et al. provides a very nice work about the  Cenomanian–Coniacian bauxite deposit in the southern Italy. The manuscript is well-written and well-organized, I don’t have too much queries about the evidences and their scientific explanations. I think this manuscript could publish just follow some minor revisions.

1.   Authors may need to add the age information of this bauxite deposit in both title and abstract. For readers’ convenience to find this key information.

2.   Please provide more details for Fig. 2 a and b, use arrows or lines to show the erosion features and ore body in photos. Besides, scale bar is needed for all photos.

3.   Texts in scale bar of Fig.3 is too small to read.

4.   Is that possible to add the values of peaks in Fig. 5 to help readers to testify your XRD explanations?

5.   Need to use wt.% instead of % for all weight precent unit for major elements. What is the unit for y axis of Fig. 7? Need to add in the figure. Same question to Fig. 8.

6.   Line 203-208, need to explain how do you calculate the Ce/Ce* and Eu/Eu*.

7.   What does “Fractal dimension” mean in subtitle 5.1? You mean the shape and size of ooids? I think you may just say microscopic feature.

Author Response

Response to Reviewer 1 Comments

Authors may need to add the age information of this bauxite deposit in both title and abstract. For readers’ convenience to find this key information.

Authors response: As suggested, the age of the studied bauxites have been added in both title and abstract.

Please provide more details for Fig. 2 a and b, use arrows or lines to show the erosion features and ore body in photos. Besides, a scale bar is needed for all photos.

Authors response: The requested information for Figure 2 has been added to the caption

Texts in scale bar of Fig.3 is too small to read.

Authors response: The scale of Figure 3 has been enlarged as suggested

Is that possible to add the values of peaks in Fig. 5 to help readers to testify your XRD explanations?

Authors response: Dear Reviewer, thank you for your suggestion, actually we decided not to add the values of XRD peaks since in this way the figure would results chaotic and confused.

Need to use wt.% instead of % for all weight percent units for major elements. What is the unit for y axis of Fig. 7? Need to add in the figure. Same question to Fig. 8.

Authors response: We indicated wt.% for major elements in the paper and, in addition, y axis of Figure 7 (wt.%) and Figure 8 (ppm) has been added accordingly.

Line 203-208, need to explain how do you calculate the Ce/Ce* and Eu/Eu*.

Authors response: The formula for the calculation of Ce/Ce* and Eu/Eu* has been added in the footnotes of Table 1.

What does “Fractal dimension” mean in subtitle 5.1? You mean the shape and size of ooids? I think you may just say microscopic features.

Authors response: We changed the subtitle 5.1 from “Fractal dimension” to “Image analysis”.

Reviewer 2 Report

1、Fractal Method is applied in the image analysis of the baxite mineralogy, which is innovative.

2、I suggest the author should change the term baxite deposite to baxite occurence since  all these deposits mentioned seems to be quite small and no systematic research had been conducted, which means  the term occurence shoud be more suited rather than deposit.

3、Another suggestion is that  geology of one of these"deposits" should be detailed if possible.

Author Response

Response to Reviewer 2 Comments

Fractal Method is applied in the image analysis of the bauxite mineralogy, which is innovative.

I suggest the author should change the term bauxite deposits to bauxite occurrence since all these deposits mentioned seems to be quite small and no systematic research had been conducted, which means the term occurrence should be more suited rather than deposit.

Authors response: Thank you for your suggestion. In the text, the term “bauxite deposit” has been changed accordingly (bauxite occurrence, bauxite bodies were used in this regard).

Another suggestion is that geology of one of these "deposits" should be detailed if possible.

Authors response: Dear reviewer, thank you for your suggestion, the geology and the formation processes of these “deposits” has been explained in the paper.

Reviewer 3 Report

See attached file, please!

Author Response

Response to Reviewer 3 Comments

Commented [ПW1]: It is need to explain! That kind, namely?

Authors response: Ca-rich matrix is related to the mineralogical composition of the matrix composed by calcite.

Line 78: Commented [ПW2]: !?

We changed to karst

Line 82: Commented [ПW3]: ? Radiolitid Limestone Fm.?

Radiolitid Limestone Fm. was added

Line 88: carsick

We changed to karst

Line 96: karsism

The term karsism is correct since it was also used by Vitale et al., 2018

Line 101: Commented [ПW3]: Need to explain the red lines!

Red lines have been explained in the Figure caption

Line 106: Commented [ПW5]: ! Radiolitid Limestone Fm.?

Fm has been added to the text

Line 152: Commented [В6]: Scales on the photos must be more!

Authors response: The scales of Figure 3 have been enlarged as suggested

Commented [ПW7]: Need to explain in “note”!

Authors response: The explanations have been added in the footnotes of Table 1.

Commented [ПW8]: Need to explain!

Authors response: The value of Sr in the sample CS2 has been explained in the text.

Commented [ПW9]: Where in table?

Authors response: “d.l.” in the Table 1 is located in the 2nd column

Line 190: Commented [ПW10]: It is necessary to sign the x-axis!

Authors response: Y axis of Figure has been added

Line 201: Commented [ПW11]: It is necessary to sign the x-axis!

Authors response: Y axis of Figure has been added

Line 203: Commented [ПW12] Where is in the table 1?

Authors response: The total REEs has been added in Table 1 as indicated

Line 204: Commented [ПW13] The host rocks of bauxites are shallow-water limestones! That is, sedimentary origin. In this case, it is necessary to normalize to shale standards (NASC, PAAS).

Authors response: Dear reviewer, in all our works on bauxite and similarly to the majority of bauxite studies, we always normalized to chondrite values. For these reasons, chondrite normalization was also preferred in order to make a comparison between the different studied bauxite deposits.

Line 207: Commented [ПW14] Anomaly variation limits and anomaly calculation formula?

Authors response: The formula for the calculation of Ce/Ce* and Eu/Eu* has been added in the footnotes of Table 1.

Line 211: Commented [ПW15] It is necessary to sign the x-axis!

Authors response: X axis has been signed as suggested

Line 211: Commented [ПW16] Must be normalized to shale standards (NASC or PAAS)! See Figs 11-12!

Authors response: As previously stated, in all our works on bauxite and similarly to the majority of bauxite studies, we always normalized to chondrite values. For these reasons, chondrite normalization was also preferred in order to make a comparison between the different studied bauxite deposits.

Line 212: Commented [B17] Need to explain positive and negative Ce-anomalies, and negative Eu-anomalies!!

Authors response: Ce and Eu anomalies have been explained in the text as suggested.

Lines 233-234 Commented [B18]  Fig. 9 - where can we see “the same emersion span of the Apennine platform”?

Authors response: This sentence refers to the fact that it has been proved that fractal dimension D values in the previously studied Campania bauxites give information on the duration of the carbonate platform emergence period. Since we found almost identical values in the Vitulano bauxites, this data could reveal that the Vitulano bauxites formed during the same emersion period.

Lines 236 Commented [B19]  It is need to explain

Authors response: The explanation of the binarization of sample image for image analysis is provided in the “Sampling and analytical methods” from Line 128 to line 131.

Lines 238 Commented [B20]  References!

Authors response: References have been added

Lines 241 Commented [B21]  References!

Authors response: References have been added

Lines 253-254 Commented [B22]  It is need to explain! That kind of process? More details!

Authors response: The period has been modified

Lines 265 Commented [B23]  ??? “stratigraphic evolution”- that is it?

Authors response: Modified into sedimentological evolution

Lines 268 Commented [B24]  Block-scheme?

Authors response: The figure caption has not been modified

Lines 269 Commented [B25]  The reference is incorrect! There is no such figure in [27]!

Authors response: The reference has been modified accordingly with [30]

Lines 271 Commented [B26]  That kind of “event”?

Authors response: The tectonic event has been reported

Line 287 Commented [B27]  Need to add references about the chemistry of rocks and bauxites areas!

Authors response: References have been added in Figure 11 and Figure 12 caption accordingly

Line 292 Commented [B28]  References!

Authors response: References have been added in Figure 11 and Figure 12 captions accordingly

Line 295 Commented [B29]  References!

Authors response: References have been added in Figure 11 and Figure 12 caption accordingly

Line 323 Commented [B30]  Need to explain positive and negative Ce-anomalies, and negative Eu-anomalies!!

Authors response: Positive and negative Ce anomalies and negative Eu anomalies have been explained in the paper as suggested.

Reviewer 4 Report

Dear Editor, the MS is very well written and in my opinion deserves publishing with minor corrections as described in detail below:

Section 5.1 Fractal dimension in bauxites: part of this belongs to the results. The authors must present the data in the Results Section not in in Discussion. In discussion only the comparison with other bauxite strata.

Fig. 10 and related discussion: in my opinion this part is very weak and not supported by the data presented in this paper. The authors did not work on the overlying strata, do not present any info and analyses, hence their figure is very stochastic and I propose to omit and keep the discussion on the formation of the bauxite and the processes during that time. 

The discussion shall start from the older processes (uplift/bauxite deposition) and end with the submerging. And focus more in the bauxite-forming period. 

Conclusions shall be re-written; now its like a repetition. They must be more concise and not descriptive.

Finally, I would propose to incorporate some comparative discussion with the karstic bauxites in nearby Greece.  

Author Response

Response to Reviewer 4 Comments

Dear Editor, the MS is very well written and, in my opinion, deserves publishing with minor corrections as described in detail below:

Section 5.1 Fractal dimension in bauxites: part of this belongs to the results. The authors must present the data in the Results Section, not in the Discussion. In discussion only the comparison with other bauxite strata.

Authors response: Dear reviewer, thank you for your suggestion. Image analysis results have been added in the Results section as suggested.

Fig. 10 and related discussion: in my opinion, this part is very weak and not supported by the data presented in this paper. The authors did not work on the overlying strata, and do not present any info and analyses, hence their figure is very stochastic and I propose to omit and keep the discussion on the formation of the bauxite and the processes during that time.

Authors response: Dear reviewer, as suggested, Figure 10 has been deleted from the paper. Figures have been re-numbered accordingly

The discussion shall start from the older processes (uplift/bauxite deposition) and end with the submerging. And focus more on the bauxite-forming period. 

Authors response: In accordance with your suggestion, in this section, we first explained the older geological and tectonic processes and then discussed the compositional features.

Conclusions shall be rewritten; now it’s like a repetition. They must be more concise and not descriptive.

Authors response: Thank you for your suggestion, the conclusions section has been rewritten in a more concise way.

Finally, I propose incorporating some comparative discussion with the karstic bauxites in nearby Greece.

Authors response: Dear Reviewer, thank you so much for your suggestion. In any case, we think it is not appropriate to make a comparison with bauxite deposits in Greece since the work focuses on Italian bauxites and, specifically, on Campania bauxite

Round 2

Reviewer 3 Report

 Fig. 9: “Binary images” - It is need to explain in the figure caption!

Fig. 10: - “Eu/Eu*” – must be indicated the normalization (to chondrite)!

             - “Limestone values are from [11]” – REE data for limestones are absent in [Mongelli et al., 2014]!   (It is shown only on fig. 9, but not in the tables! “Bedrock limestones” have been normalized to chondrite?)

My recommendation - to remove it from Fig. 10!

Fig. 11: - “Eu/Eu*” – must be indicated the normalization (to chondrite)!

                -“Upper continental crust” - references!

Author Response

Response to Reviewer 2 Comments

Fig. 9: “Binary images” - It is need to explain in the figure caption!

Authors response: Thank you for your comment. The explanation of the binarization of images has been added in the caption accordingly.

Fig. 10: - “Eu/Eu*” – must be indicated the normalization (to chondrite)!

             - “Limestone values are from [11]” – REE data for limestones are absent in [Mongelli et al., 2014]!   (It is shown only on fig. 9, but not in the tables! “Bedrock limestones” have been normalized to chondrite?)

My recommendation - to remove it from Fig. 10!

Authors response: Thank you for your suggestions. Chondrite normalization has been reported in the figure and Limestone value has been deleted from Figure 10 as suggested.

Fig. 11: - “Eu/Eu*” – must be indicated the normalization (to chondrite)!

                -“Upper continental crust” - references!

Authors response: Thank you for your comment. Chondrite normalization has been reported in the figure and UCC reference has been added as indicated.
